



# Assessing the drift of Fish Aggregating Devices in the tropical Pacific Ocean

Philippe F.V.W. Frankemölle[1,2], Peter D. Nooteboom[1,3], Joe Scutt Phillips[4], Lauriane Escalle[4], Simon Nicol[4,5], and Erik van Sebille[1,3]

[1]Institute for Marine and Atmospheric Research (IMAU), Department of physics, Utrecht University
[2]Marine and Fluvial Systems, Department of Engineering Technology, University of Twente
[3]Centre for Complex Systems Studies, Utrecht University
[4]Oceanic Fisheries Programme, Pacific Community
[5]Centre for Conservation Ecology and Genomics, Institute for Applied Ecology, University of Canberra

**Correspondence:** Philippe Frankemölle (p.f.v.w.frankemolle@utwente.nl)

**Abstract.** The Tropical Pacific Ocean is characterized by its dominant zonal flow, strong climate dependence on the El Niño Southern Oscillation (ENSO) and abundant tuna stocks. Tuna fisheries in the West and Central Pacific Ocean accounted for 55% of world-wide tuna catch in 2019 and are one of the main sources of income in many Pacific island nations. One of the dominant fishing methods in this region relies on the use of drifting Fish Aggregating Devices (dFADs): rafts with long underwater appendages (on average 50m deep) that attract and aggregate fish. Although currents such as the North Equatorial Countercurrent (NECC) and South Equatorial Current (SEC) in the tropical Pacific Ocean vary strongly with ENSO, little is known about the impact of this variability in flow on dFAD dispersion. In this study, virtual Lagrangian particles are tracked for the period 2006 to 2021 over the domain in a 3D hydrodynamic model and are advected in simulations with only surface flow as well as simulations using a depth-averaged horizontal flow over the upper 50 meters. The particle trajectories are used to determine zonal displacements, eddy-like behaviour and ENSO variability for drifters that are subjected to either surface or depth-averaged currents. It was found that virtual particles that are advected by only surface flow are displaced up to 35% farther than those subjected to a depth-averaged flow, but no other major differences are found in dispersion patterns. Strongest correlations between ENSO and dFAD dispersion for the assessed variables were found in the West Pacific Ocean, with Pearson correlation coefficients up to 0.59 for dFAD displacement. Connections between ENSO and eddy-like behaviour were found in the western part of the SEC, indicating more circulation and meandering during el Niño. These findings may be useful for improving sustainable deployment strategies during ENSO events, and understanding the ocean processes driving the distribution of dFADs.

## 1 Introduction

In 2019, tuna fisheries in the West and Central Pacific Ocean (WCPO) accounted for 55% of worldwide tuna catch (Williams and Ruaia, 2020). One of the dominant fishing methods, utilized by industrial purse seining fleets, relies on the usage of Fish Aggregating Devices (FADs) (Davies et al., 2014). These devices aggregate fish to their location, making it easier to catch large



quantities of tuna during one fishing event. These FADs can be grouped in two categories: anchored and drifting FADs (Leroy et al., 2013), where the former are generally used by coastal fleets and artisanal fishers, and the latter (dFADs) are generally used by distant-water and industrial fleets and will be the focus of this research.


In their basic form, dFADs are floating, satellite-tracked bamboo rafts with appendages underwater that can reach to average depths of 50m or more (Fonteneau et al., 2000). Due to these appendages, dFADs move slower than surface drifters (Imzilen et al., 2019). The concept of a dFAD is based around the behavioural tendency of pelagic fish, such as tuna, to associate with floating objects (Fréon and Dagorn, 2000; Castro et al., 2002).


Most technological advances, more often than not kept as trade secrets, share a common goal: increasing dFAD efficiency (Fonteneau et al., 2000). These advances are essentially a double-edged sword. On one hand, they can help make the fishing more cost-effective and reduce carbon emissions for a purse seining fleet, which is a highly mobile and dynamic fishery. On the other hand however, it could lead to overfishing of stocks, unnecessary by-catch and damage to the ecosystem (Dagorn

et al., 2013). Both papers by Dagorn et al. (2013) and Bailey et al. (2013) point out that with proper management these devices can be used as a safe and sustainable fishing method, and building knowledge on dFAD dynamics is important for sustainable management. Similarly, the marine pollution and damage to coastal habitats from abandoned, lost or otherwise discarded dFADs is of increasing concern (Escalle et al., 2019). Here we study these oceanographic dynamics of dFAD dispersion in the tropical Pacific Ocean, where most tuna is caught within the West Pacific warm pool (Lehodey et al., 1997) and which

influences dFAD-tuna interactions (Nooteboom et al., 2023; Pérez et al., 2022).

Presumably, three main surface currents influence dFAD dispersion in the tropical Pacific Ocean: the North Equatorial Current (NEC), the South Equatorial Current (SEC) and the North Equatorial Counter Current (NECC). The first two currents flow eastward under normal conditions and are part of the two basin-wide Pacific circulation gyres. The latter current is a westward

flow that transports water from the warm pool to the Eastern Pacific Ocean (EPO) and separates the NEC from the SEC (Zhao et al., 2013). A general map showing these currents is given in Fig. A1 in the appendix. The NEC and SEC are close to the ocean surface, and dominated by the trade winds which also blow in the eastward direction. While these currents are well studied, the effect of their inter-annual variability due to the El Niño Southern Oscillation (ENSO) cycle on dFAD drift remains less characterised. In particular, accumulation patterns and dFAD drift are more variable in the western warm pool where the

oceanography is complex and influenced more by ENSO phase, than in the EPO (Escalle et al., 2021c; Lopez et al., 2020).

During a positive ENSO phase the tradewinds weaken or even reverse, changing large scale circulation patterns (Horel, 1982) and shifting the warm pool in the zonal direction (Picaut et al., 1996). During an el Niño event, the SEC becomes weaker and the NECC becomes stronger (Wyrtki, 1975). Not only does ENSO affect the large-scale circulation, it also influences

mesoscale eddies in the tropical Pacific Ocean. An important mechanism in this study region is the barotropic shear introduced between the westward SEC and the eastward NECC (Willett et al., 2006). This generates tropical instability waves - westward



travelling SST anomalies - and tropical instability vortices - anticyclonic eddies also travelling in the westward direction. This velocity shear weakens during an el Niño event, whereas this shear is strengthened during la Niña years resulting in stronger tropical instability waves and vortices in the latter case (Wang and Fiedler, 2006).


Although considerable research has been done on the el Niño phenomenon and even on its relation to tuna catch (e.g., Lehodey et al., 1997; Lehodey, 2001; Lu et al., 2001; Kumar et al., 2014; OFP-SPC, 2021), the impact of ENSO on the temporal variability of dFAD dispersion in the tropical Pacific Ocean is unknown. The increased attention on the number and distribution of dFADs has raised concerns on the impacts of not only dFADs actively used by fishers, but also those drifting

outside of the equatorial fishing zone which subsequently cause pollution and stranding (Escalle et al., 2019). Spatial management has been suggested as one solution to this problem (Imzilen et al., 2021), but this requires a strong characterisation of ocean flow and its impacts on dFAD dispersion in a region. In parallel, biodegradable dFADs are now being developed (Escalle et al., 2022a) as a possible mitigation against the marine pollution caused by conventional structures, although their degradation is highly variable (Moreno et al., 2023) and likely to be non-linear. Regardless, although all lost dFADs may retain

their entire subsurface structure, many are found as simply rafts or even GPS buoys, having presumably broken apart following long drift-times or rough ocean conditions (Escalle et al., 2022b). These remnants experience more surface-driven currents than when they were originally drogued at depth, further complicating spatial management and predictions of high density areas and corridors of loss.

Despite these complications, the drift trajectories of dFADs clearly have the potential to supplement the data provided by drifters for assimilation into ocean circulation models or other applications (Imzilen et al., 2019). It is estimated that 35 to 65,000 dFADs are deployed in the Pacific Ocean each year (Escalle et al., 2021b; Lopez et al., 2020) Understanding not only how dFAD drift-profiles may change through their lifetime, but also where and how their drift patterns may differ during their operational lifetime is a critical step towards such an objective.


Here we compare dFAD drift with virtual particle drift solely due to surface currents, given that dFADs are drogued during their operational lifetime, and thus influenced by deeper currents up to 50 meters. Additionally, we study the correlation between ENSO variability and dFAD displacement, dispersion and metrics that quantify the 'loopiness' of their trajectories (e.g. Doglioli et al., 2006). This is done by seeding Lagrangian particles in the Pacific Ocean from 15°S to 15°N in a 3D

hydrodynamic model using the horizontal flow at multiple depths.

## 2 Methods

### 2.1 Data-sets

*Ocean Currents*





We used velocity fields from the non-profit organisation Mercator Ocean International (MOi). The horizontal velocity fields
at all depths up to 50m were obtained from the MOI_GLO12_WEEKLY_run_for_DAILY_FORECAST product (Mercator
Ocean International, 2021). These data are the daily-mean output of a 3D global ocean model assimilated with satellite data
and *in-situ* observations, using the GLORYS assimilation scheme. The fields have a horizontal resolution of 1/12° , with 50
layers, and are available on an Arakawa C-grid, without interpolation. It is known that the tropical SEC is slightly too strong in
the MOi data when compared to *in-situ* 15m drogued drifters (Lellouche et al., 2021), which may result in an overestimation
of dFAD displacement and loopiness for virtual particles released in the SEC. Currently used data spans from 2006-10-11 to
2021-12-31.

*Sea Surface Temperature*

SST data is taken from the NOAA Physical Sciences Laboratory. The Niño 3.4 SST Index is used (National Ocean and At-
mospheric Administration, 2022), which is derived from SST values from the Met Office's Hadley Centre Sea Ice and Sea
Surface Temperature data set (HadISST1). For data past 1982, these values are a mixture of *in-situ* measurements and satellite
data (Rayner et al., 2003; National Centre for Atmospheric Research, 2022). The NOAA data-set gives the spatial mean of the
SST over the Niño 3.4 region (from 5°S to 5°N and 170° to 120°W (Kug et al., 2009)). These data currently span from 1870
to April 2022 on a one month interval. To transform these data into the Niño 3.4 index the monthly climatology from 1992 to
2021 is removed.

## 2.2 Virtual Particle Simulations

The simulations are done using the Lagrangian framework Parcels (Delandmeter and Van Sebille, 2019). The simulation lasts
over the entire duration of the MOi data-sets. Trajectories are outputted with a daily temporal resolution. Particles are in-
tegrated forwards in time with time-steps of 6 hours using only a kernel for two-dimensional advection. This is the limit at
which most particles do not travel more than 1 grid cell (1/12°) per time step, satisfying the Courant-Friedrichs-Lewy condition.

Every tenth day in the simulation, 17,016 new particles are released from 15°N to 15°S and 140°E to 75°W, with a fixed
0.5° spacing. Particles are deleted after 30 days due to computational limits. Vertical velocities are not taken into account. Two
different configurations are used to simulate virtual particle drift: (1) using a depth averaged flow over the upper 50 meters,
to account for dFAD appendages; and (2) using surface flow only;. As not all depth layers in the data are equally wide, each
layer's contribution to the depth-averaged velocity is weighted depending on the layer thickness.

## 2.3 Data Analysis

### 2.3.1 Variables

*Displacement distance, travel distance and distance ratio*

We use the displacement distance to quantify the 'straightness' of a trajectory. Displacement distances are calculated using the



Haversine distance: the formula for calculating the shortest distance between two points on earth. By applying this equation between every subsequent position in a trajectory and taking the sum of all of these small distances, an approximation of the travel distance is made. This method becomes more inaccurate if the temporal resolution becomes lower. The distance ratio is obtained through dividing the displacement distance by the travel distance. Ratios closer to 1 imply straight trajectories, whereas ratios closer to 0 imply circulation to the same location.

*Loopiness*

Loopiness can be used to quantify eddy-like behaviour. To obtain the loopiness of a trajectory, first the bearing (angle clockwise from the north) between each point in the trajectory is calculated. Thereafter, the smallest difference between those angles is calculated and those differences are summed up, assigning a loopiness value to a trajectory. Here, outliers occur when an angle difference comes close to plus or minus $\pi$, as this minor difference strongly effects whether the loopiness value increases or decreases. The time step here is sufficiently small that this is a very rare occurrence, such that it does not influence the statistical analysis.

### 2.3.2 Analysing ENSO variability

To study variability of dFAD dispersion in relation to ENSO composite images of positive and negative ENSO phases are created, and variables are correlated to the Niño3.4 index, where the time assigned to a particle trajectory is defined as the time at which a particle is seeded into the domain. Correlations between variables and the Niño3.4 index are done with monthly data. Note that every tenth day new particles are released on the same location, hence multiple values for displacement and loopiness are observed each month. A monthly mean of these values is taken, but due to the irregular time span of months (28-31 days) these monthly averaged values consist of 2,3 or 4 data points.

An ENSO event in this research is defined as a deviation of 1°C or higher in the Niño3.4 index that last three months or longer. Within the simulated temporal period there have been two el Niño events satisfying these criteria (2009 and 2015); there have been 3 la Niña events (2008, 2010 and 2020). In the data analysis, these ENSO events are used to create composite figures for the distance ratio and these specifications are not used for the correlation analyses.

## 3 Results

### 3.1 Displacement

Variability of the (depth-averaged) flow field in the tropical Pacific Ocean with respect to ENSO is more dominant in the zonal direction than in the meridional direction, which by extension also holds for dFAD displacement. Fig. 1 shows the correlation (top figure) and linear regression coefficient (bottom figure) between dFAD displacement after 1 month and the ENSO index. Overall, the largest areas with significant correlations are centered around the West and Central Pacific Ocean.





The strongest variability is shown at the North Equatorial Countercurrent, with a maximum correlation coefficient of 0.59, from where simulated dFADs can end up 293±30km further away per unit of ENSO change. This is a considerable change, considering the average dFAD released from this location travels roughly 1000km in the simulation. This observation of dFADs

drifting further east during an el Niño event is in accordance with indications of increased strength of the NECC during el Niño (Zhao et al., 2013). The striped pattern surrounding the NECC could possibly be due to the southward shift in the NECC during el Niño and the northward shift during la Niña, or due to stronger eddies in the NECC during el Niño reducing dFAD displacement. Whether this pattern is actually an ENSO phenomenon is unclear. The South Equatorial Current weakens during a warm event (Wyrtki, 1975), offering an explanation for the strongly negatively correlated blue area, indicating that dFADs

originating here are less displaced.

## 3.2   Mesoscale phenomena

Not all behaviour can be attributed solely to large scale zonal flow, as mesocale phenomena also impact dFAD trajectories. Besides only looking at how far dFADs have displaced, the full trajectory can also be taken into account. In Fig. 2 the ratio between displacement distance and the travel distance is given, as well as composite images for the el Niño and la Niña events.

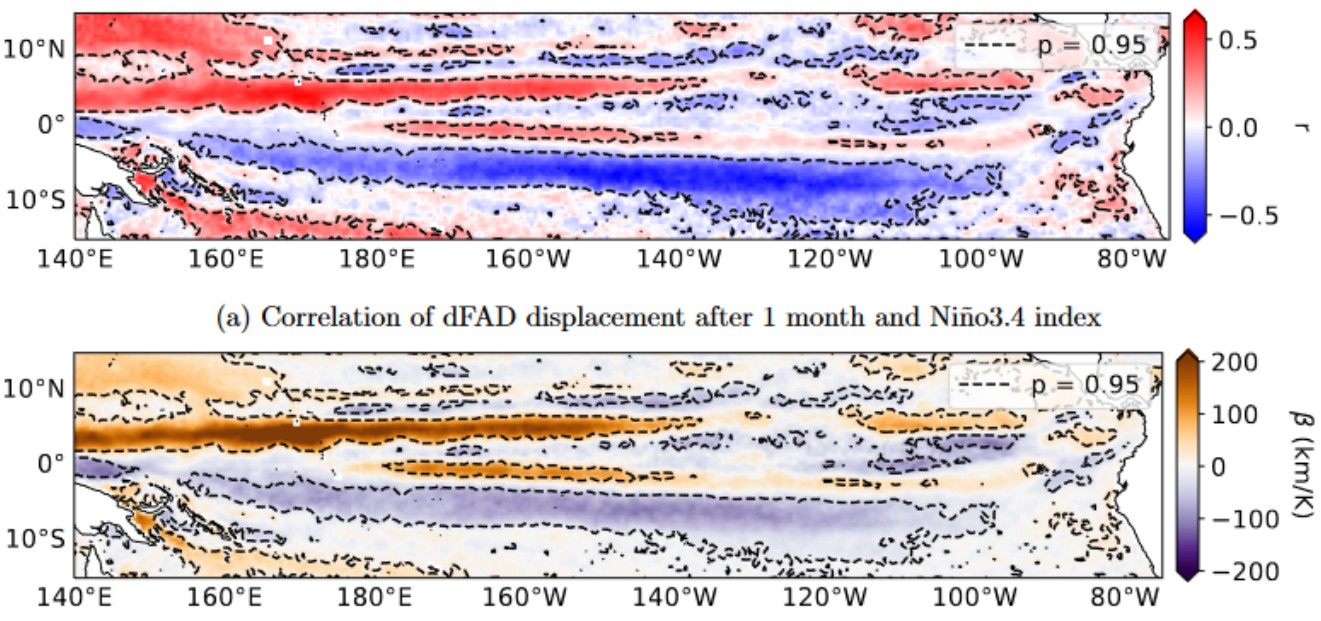

(a) Correlation of dFAD displacement after 1 month and Niño3.4 index

(b) Regression coefficient of dFAD displacement after 1 month as a function of the Niño3.4 index

**Figure 1. (a)** Correlation between dFAD displacement after 1 month and the monthly Niño3.4 index. Positive correlation are shown in red, whereas blue colours show a negative correlation. The dotted black lines mark regions where correlations are significant (probability values larger than 95%). **(b)** Linear regression coefficient between dFAD displacement after 1 month and the monthly Niño3.4 index. Units are in $kmK^{-1}$.




When comparing la Niña events to averaged conditions, distance ratios tended more towards the extremes in the open ocean. This means that straight trajectories appear even straighter and wavy trajectories meander or spiral even more. This supports the hypothesis that la Niña events mainly amplify the usual flow pattern. In the case of an el Niño, a large difference occurs in the western part of the SEC. Very low ratios are observed in the same region as in Fig. 1 where the displacement was strongly negatively correlated with the ENSO index. This new information suggests that in this region, where many Pacific

island nations are located, the decrease in dFAD displacement during el Niño events is also due to eddy-like behaviour. The distance ratio in Fig. 2b north and south of the NECC is also very low, which may be attributed to the strong meandering this current experienced in 2010 (Zhao et al., 2013). However, it is important to keep in mind that these composites are comprised

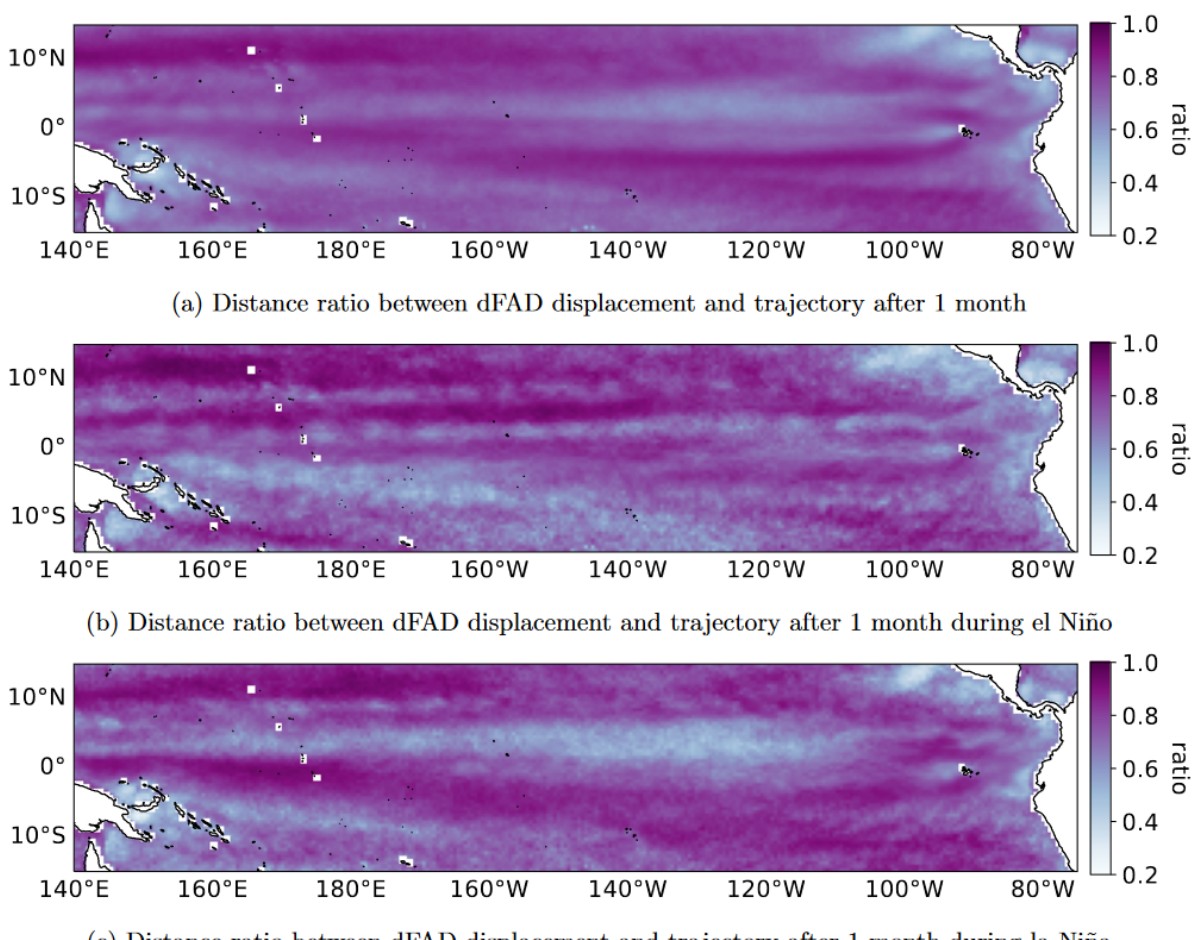

(a) Distance ratio between dFAD displacement and trajectory after 1 month

(b) Distance ratio between dFAD displacement and trajectory after 1 month during el Niño

(c) Distance ratio between dFAD displacement and trajectory after 1 month during la Niña

**Figure 2.** Composite figures showing the distance ratio between dFAD displacement and the full trajectory after one month of travel during **(a)** all 15 years, **(b)** el Niño events and **(c)** la Niña events. Values closer to 1 (in dark purple) indicate nearly straight trajectories, whereas values closer to 0 (in white) indicate stronger circulation.





of respectively only two and three events. This means that differences between composite figures do not exclusively showcase ENSO variability, but may also demonstrate noise.


Whereas the distance ratio measures how straight particle trajectories are, the loopiness parameter quantifies the total change in direction by taking the sum of all angles during a one month trajectory. The modelled loopiness is shown in Fig. 3a. The most notable feature in this figure is the near anti-symmetry between the Northern and Southern Hemisphere: dFAD loopiness generally is clockwise in the north, but counterclockwise in the south. This can almost certainly be traced back to the direction

of the Coriolis acceleration on each Hemisphere. A further feature in this figure are the very low loopiness values near the equator. This may indicate that dFAD trajectories near the equator go straight, however it is also plausible that this feature is the result of the positive and negative loopiness values corresponding from different langrangian trajectories averaging each other out.

Figs. 3b and 3c show a large area slightly south of the equator where loopiness is significantly negatively correlated to the Niño3.4 index. Judging from the minimum regression coefficients during el Niño events, dFADs seeded in this region can experience more than 20% change in loopiness per unit Kelvin. This region coincides with the region where the distance ratio is close to zero, supporting the observation that during el Niño events, eddy-like behaviour in the southern part of the domain becomes more prevalent. However, a negatively correlated patch is also observed north-east of the domain, where the NECC

and the SEC shear. Together with the larger distance ratios observed in the East Pacific Ocean in Fig. 2b relative to Fig 2c, this indicates more eddy-like behaviour during la Niña and less during el Niño in this region. These observations seem in agreement with the theories that tropical instability vortices (which grow between 3°N and 8°N in the East Pacific Ocean) are less active during el Niño years (Yu and Liu, 2003).

### 3.3 Depth-averaged flow versus surface flow

The appendages of a dFAD are meant to slow down the dFAD due to the slower moving currents below the surface. In theory, this opens up possibilities for new interactions that are unseen if only surface currents are considered, which may affect the drift of dFAD rafts after their eventual break down and sinking of their negatively buoyant appendages.

Fig. 4 shows the differences between the two simulations. The displacement figures show mainly the same pattern, though

with displacements up to 35% higher over the entire domain when only accounting for surface flow. These higher displacements for surface particles are most likely due to the absence of slower subsurface currents that slow down the dFAD particles. Distance ratios during el Niño are nearly identical in both cases. There is however a notable difference with higher values on the south-east of the domain, where particles appear to travel in generally straighter lines in the simulations with just surface flow. Similar differences are not found in other comparisons. Lastly, the regression coefficients of loopiness with respect to

the Niño3.4 index are shown in Figs. 4e and 4f. Patterns are similar, with the only difference that the drogued particles in the



**Figure 3. (a)** dFAD loopiness in radians after 1 month. Red values mark clockwise motion and blue values mark counterclockwise motion. **(b)** Correlation between dFAD loopiness after 1 month and the monthly Niño3.4 index. Positive correlations are shown in red, whereas blue colours show negative correlations. The dotted black lines mark regions where correlations are significant (probability values larger than 95%). **(c)** Linear regression coefficient between dFAD loopiness after 1 month and the monthly Niño3.4 index. Units are in $\mathrm{radK}^{-1}$, where K is the unit of the ENSO index.

strongly negatively correlated area experience weaker variability. A likely explanation for this behaviour is that particles with lower velocities travel shorter distances and thus make less loops, which by extension leads to weaker ENSO variability.



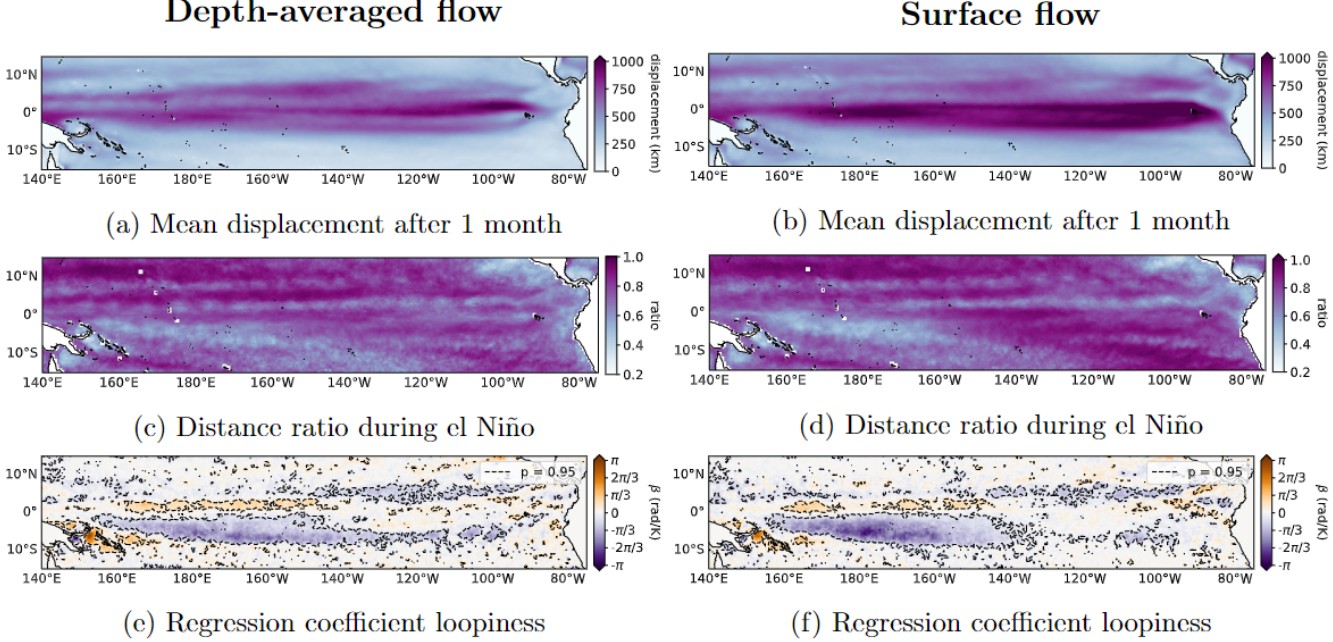

**Figure 4.** Comparison between virtual particle simulations with depth-averaged flow (left) and only surface flow (right). **(a,b)** Mean particle displacement after 1 month of travel. **(c,d)** Distance ratio between displacement and travel distance after 1 month of travel during el Niño. **(e,f)** Linear regression coefficient between dFAD loopiness after 1 month and the monthly Niño3.4 index.

## 4   Discussion and outlook

In this paper, we analysed the drift of dFADs in the tropical Pacific Ocean, using Lagrangian particle tracking. To estimate the
effect of their 50m long appendages, we applied two types of simulations. The first was performed using a depth averaged flow
over the upper 50m, weighed by layer depth, and the second used only surface flow to advect particles. Using this method, we
found that the virtual particles advected by only the surface flow can be displaced up to 35% more than those subjected to the
depth-averaged flow and have a stronger loopiness variability with respect to ENSO, but no substantial differences between the
two simulations were found in patterns across the domain of the tropical Pacific Ocean.


Regions identified in this paper as having the highest loopiness and lowest displacement, the south-western part of the WCPO
(Papua New Guinea, Solomon Islands, Tuvalu, Kiribati Gilbert Islands), the south-eastern EPO and the northern part of the
EPO between 3°N and 8°N, are known areas of dFAD aggregations (Escalle et al., 2021c; Lopez et al., 2020). The zone along
the equator, characterised with very high displacements, are known high-speed drift areas of dFADs, where most deployments
occurred in the EPO, before dFADs are brought to each side of the equator by divergent currents and they either accumulate in
eddies in the northern hemisphere or they are transferred towards to southern WCPO in the southern hemisphere (Lopez et al.,





2020).

In some cases, simulations can be directly compared to observations provided by fisheries on dFAD densities (Escalle et al.,
2021a; Scutt Phillips et al., 2019), but these data on dFAD distributions do not cover the entire domain studied in this paper, nor
are they publicly accessible. Such data could be used to potentially improve the model, for example improving the calculation
of the depth averaged velocity field so that each depth layer is weighed depending on the actual shape of a dFAD instead of
only the depth of a layer. Furthermore, scenarios that vary the depth integrated velocity field over individual particle drift-time
could be undertaken, adjusting the weight given to each depth layer over time as to simulate the biodegradable aspect of dFADs
(appendages becoming shorter over time).

Correlations between dFAD displacement and the Niño3.4 index were very high in some locations, up to a maximum of 0.59
in the NECC, indicating strong ENSO variability. Overall, dFADs tended to be displaced more during an el Niño if seeded in
the NECC and less if seeded in part of the SEC below the equator, which can be attributed to the respective strengthening and
weakening of the currents during the positive ENSO phase. These two areas with higher correlation between dFAD displace-
ment and ENSO, correspond to areas with lower or higher dFAD aggregation during el Niño, relatively. dFAD fishing in the
WCPO NECC is generally low, and it decreases even further during el Niño, with the whole fishery moving East (Williams
and Ruaia, 2020). The WCPO SEC, indicates negative correlation between dFAD displacement and ENSO, corresponding to
an area with higher dFAD fishing effort during el Niño (Williams and Ruaia, 2020). The mechanisms behind the observed
response of the WCPO tuna fishery to el Niño (Williams and Ruaia, 2020; Lehodey et al., 1997; OFP-SPC, 2021) may be
related to these same drivers of change in dFAD drift as a result of ocean flow. The high FAD density areas in the southern
WCPO (Escalle et al., 2021c) might extend or move east during an el Niño event, although this could not be verified due to
the lack of availability of observed trajectories covering this region. Different patterns in displacement, eddy ratio, loopiness
and pathways often follow zonal lines, as the dominant currents in the open ocean of this domain travel mainly in the zonal
direction. Dispersion in the zonal direction may however be overestimated in this research as the tropical SEC is too strong in
the MOi data when compared to *in-situ* 15m drogued drifters (Lellouche et al., 2021).

In the EPO, stronger loopiness was observed during a la Niña event between the NECC and SEC. This loopiness mostly dis-
appeared during el Niño events. A possible explanation could be that during la Niña stronger generation of tropical instability
vortices takes place, due to to higher velocity shear between the eastwards and westward currents. In the WCPO, a region of
stronger eddy-like behaviour was found in the SEC, that became stronger during el Niño due to decreased distance ratios and
increased loopiness. The mechanisms behind this behaviour, either of physical nature or artifacts from the strong SEC in the
MOi data, require further research. The main findings and reasonings of this research are summarised in Fig. 5.

In determining the variability of dFAD dispersion with respect to ENSO, correlations were significant, but only a few ENSO
events were taken into account. The results consist of data with two el Niño and three la Niña events (one of which only lasted





three months above 1°C). As the currently used data sets are still being expanded upon for the foreseeable future, more events may eventually be captured and analysed leading to even more robust results.

Moreover, the phase of the ENSO event is mostly ignored throughout this paper. In some cases, ENSO related variability of Pacific Ocean currents distinguishes between the developing stages and mature stages of an el Niño/la Niña event (e.g. Wu et al., 2016; Zhai and Hu, 2012; Zhao et al., 2013; Tan and Zhou, 2018) and the delay of its effects depending on the location. In this paper, where most calculated variables in time series data are already averaged or summed over the time of a month, there is a 1 month phase lag within the calculation of one value assigned to a trajectory. Hence, it is arbitrarily chosen as the

time of dFAD deployment. The effects of phase lag of calculated values such as displacement, loopiness and distance ratio with respect to the Niño3.4 index have not been investigated here.

       Only a fraction of the amount of data MOi has to offer has been used in this study, whereas their products have a lot

more to offer for further research regarding dFAD drift. Future research could also focus on the actual link to tuna catch or effectiveness of certain release locations. Moreover, it could be interesting to track bio-activity along the dFAD trajectory, such as chlorophyll-a concentrations, to see if and how dFADs, released in particular areas, travel through nutrient-rich areas. Furthermore, we used the distance ratio and loopiness to quantify whether dFADs appear to be travelling in an eddy, classifying it as 'eddy-like behaviour'. Such data can be further built upon by tracking the along trajectory eddy diffusivity, in order to

determine the mechanisms behind specific types of trajectories.





**Figure 5.** Diagram showing the general findings and considerations of this paper. Arrows mark how findings and locations influence each other. Dashed arrows denote possible connections.



## Appendix A

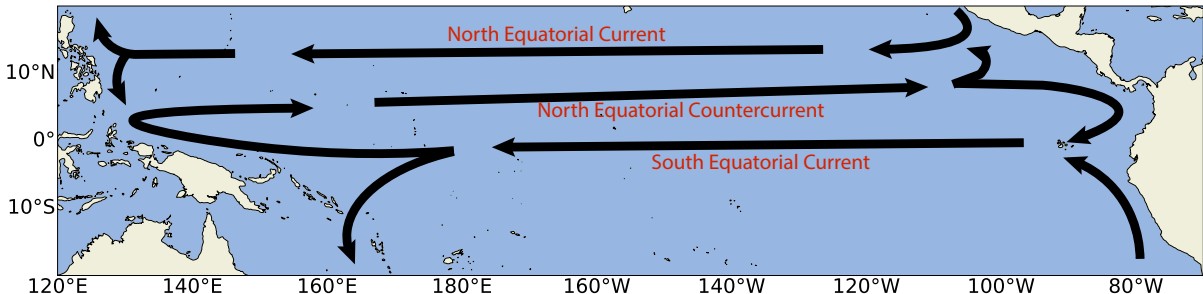

**Figure A1.** General overview of currents in the tropical Pacific Ocean. The three main currents in the open ocean are the North Equatorial Current, the North Equatorial Countercurrent and the South Equatorial Current.



*Code availability.* https://doi.org/10.5281/zenodo.7623995

*Author contributions.* PN and EvS had a supervisory role in the design and carrying out of the research. PF designed the research, developed the model code and performed the simulations. PF prepared the manuscript with contributions from all co-authors.

*Competing interests.* EvS is a member of the editorial board of Ocean Science. The peer-review process was guided by an independent editor, and the authors have also no other competing interests to declare.

*Disclaimer.* This publication was produced with the financial support of the European Union and Sweden. Its contents are the sole responsibility of the authors and do not necessarily reflect the views of the European Union and Sweden.

*Acknowledgements.* Funding was provided by the Western and Central Pacific Fisheries Commission (WCPFC Project 42) and the European
Union "Pacific-European-Union-Marine-Partnership" Programme (agreement FED/2018/397-941).





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
