# Peer review of "Assessing the drift of Fish Aggregating Devices in the tropical Pacific Ocean"

_EGUsphere, 2023_

## Author Response (AR1)

**Assessing the drift of Fish Aggregating Devices in the tropical Pacific Ocean: author's response**

The response to reviewers #1 and #2 is nearly identical to our initial response to the individual reviews. However, line numbers and wordings may differ slightly compared to the first response. Pages 12 and 13 provide a comprehensive list of all relevant changes to the manuscript.

**Response to first reviewer**

*In this manuscript the authors present the analysis of an extensive dataset of Lagrangian trajectories simulated using modelled velocity fields across the Pacific Ocean. The authors also propose a connection between simulations and real dispersion of FADs across the basin. Overall the work is interesting but I believe that some critical points should be addressed before publication. Please find below my argumentations that hopefully will be useful for the authors:*

We thank the reviewer for their careful and useful comments. Below, we detail how we address all of them.

**Main points:**

*1) The FADs dataset is mentioned several times across the paper but it is indeed poorly described and characterized. It is not even clear if and how the authors really use FADs observed trajectories or not. How the simulations presented can be tested against real FADs pathways?*

Perhaps it was unclear in the initial manuscript that our analysis is based purely on trajectories obtained from numerical simulations. We did not employ actual dFAD data, which is not openly available in the Tropical Pacific. In the introduction, we use FADs to frame our research question as a motivation to assess hydrodynamic connectivity in the Tropical Pacific. We have now clarified this by specifically referring to '*virtual dFAD particles*' throughout the manuscript.

*2) A key point is that a FAD is a large floating object with a significant mass and complex hydrodynamical proprieties. However the authors claim that simulated numerical passive tracers (i.e. ideal point-like, massless particles) can effectively describe the dynamics of FADs. This is a very difficult to support assumption. More generally, I would say that the connection with FADs dynamics is pretty weak. Either the authors provide more elements to support the relevance of their simulation for*

addressing real FADs dynamics or they shift the focus on general Lagrangian dispersion patterns in the region.

The reviewer is right that dFADs are large floating objects and thus their hydrodynamic properties are different from other objects. On the scale of the global ocean, the FADs do not have their own inertia (the Stokes number is very low), see also Van Sebille et al (2020). Furthermore, it is not true that we consider the dFADs as point-particles. In our simulations, we integrate the velocity over the full depth range (50m) of a dFAD.

In lines 117-120, of the method section, we have added the following:

*"This means we do not account for the mass and small-scale hydrodynamic properties of dFADs in this study. On the scale of the global ocean this assumption holds, as the dFADs have a very low Stokes number and as such do not have their own inertia (Van Sebille et al, 2020)."*

3) The authors present some interesting statistical patterns of the loopiness metric across space and time. However they do not discuss in depth how such patterns could be related to studies of eddies polarity statistics in the same basin present in the literature (e.g. Abernathey, R., & Haller, G. (2018). Transport by lagrangian vortices in the eastern pacific. J. Physical Oceanography, 48, 667–685).

Literature indicates that Tropical Instability Waves (TIW) are a cause of eddies in the South Equatorial Current (SEC), because TIWs are characterized by large scale meandering of the SEC (as observed originally by Düing et al.: 1975). TIWs are stronger during a la Niña event (Imada and Kimoto 2012; Yu et al 2003), and are not as pronounced south of the equator (Xue et al., 2020).

Indeed, the negative correlation between ENSO and loopiness in the EPO indicates increased eddy-like behaviour during la Niña. To clarify this, we now added lines 206-209 (results):

*"This observation is in agreement with the theories that tropical instability waves are stronger in this region during a la Niña event (Yu et al., 2003; Imada and Kimoto, 2012) and that the associated tropical instability vortices (which grow between 3$^o$N and 8$^o$N in the East Pacific Ocean) are less active during el Niño years (Yu et al., 2003)."*

In the Western and Central Pacific SEC region, however, we observe increased loopiness and decreased distance ratios in the negatively correlated loopiness area. We could not find literature which links ENSO and eddy dynamics in this region. For example, the research of Abernathey et al. (2018) shows that in the tropics of the eastern Pacific few eddies appear per square degree of latitude/longitude per year, but do not offer much insight with respect to the spatial and temporal distribution of eddies in the tropics; the paper focusses more on the subtropics, which is the case for most studies related to eddy polarity statistics in this region. As such, we changed the following to the discussion in lines 266-273:

*"In the EPO, stronger loopiness and lower distance ratios between the NECC and SEC (north of the equator) were observed during colder conditions. This coincides with the region where during la Niña, stronger generation of tropical instability vortices takes place, due to higher velocity shear between the eastward and westward currents (Yu et al., 2003; Imada and Kimoto, 2012). In the WCPO, a region of stronger eddy-like behaviour was found in the SEC, that became stronger during warm conditions, with decreased distance ratios and increased loopiness. South of the equator, weak TIW patterns exist (Xue et al., 2020) but they should have an opposite effect on our eddy statistics, compared to our observations, which suggests TIWs are not the driving mechanism behind these patterns in the WCPO SEC. The mechanisms behind this behaviour require further research, as most research on eddy polarity statistics in this region do not focus on the tropics."*

Zhao et al. (2013), states that during the 2009-2010 el Niño event, the North Equatorial Counter Current (NECC) was displaced southwards during the warming stages and that its intensity (flow velocities) increased. Zhou et al. (2021) shows that the same happened during the strong 2014-2016 el Niño event, though this study focused mainly on the origin of the NECC in the far west (outside our region of interest). Both argue that the increased velocity shear somewhat leads to increased meandering and recirculation in this region. In our data, the flow of the NECC intensifies and particle trajectories follow straighter trajectories during positive ENSO events in the NECC. In our response to the specific comment on line 156 below, we discuss how we implement this in the manuscript.

[1]Düing, W., Hisard, P., Katz, E., Meincke, J., Miller, L., Moroshkin, K. V., ... & Weisberg, R. (1975). Meanders and long waves in the equatorial Atlantic. *Nature*, *257*(5524), 280-284

[2]Zhao, J., Li, Y., & Wang, F. (2013). Dynamical responses of the west Pacific North Equatorial Countercurrent (NECC) system to El Niño events. *Journal of Geophysical Research: Oceans*, *118*(6), 2828-2844.

[3]Zhou, H., H. Liu, S. Tan, W. Yang, Y. Li, X. Liu, Q. Ren, and W. K. Dewar, 2021: The Observed North Equatorial Countercurrent in the Far Western Pacific Ocean during the 2014–16 El Niño. *J. Phys. Oceanogr.*, **51**, 2003–2020

[4]Xue, A., Jin, F. F., Zhang, W., Boucharel, J., Zhao, S., & Yuan, X. (2020). Delineating the seasonally modulated nonlinear feedback onto ENSO from tropical instability waves. *Geophysical Research Letters*, *47*(7), e2019GL085863.

**Specific comments on the manuscript:**

(a) line 80-81 : not really clear how the authors compare FADs drift and virtual particles trajectories

In this research, the FADs are virtual as well, which we have clarified now by referring to 'virtual dFAD particles' throughout the manuscript. We compared the virtual FAD drift and virtual particles advected only due to surface currents by comparing in both cases the statistics of our metrics (averages, correlations and linear regression coefficients). In lines 195-207 we describe the comparison of some of these metrics. However, the main result here is that the virtual FAD drift and the drift of virtual particles advected only due to surface currents, were quite similar in their spatial structure, but mainly showed differences amplitude-wise.

(b) line 111-112 : Why particle are removed? It seems that the number of particles is far smaller than numbers that can be numerical unfeasible.. Which is the real numerical limitation here? A few thousands of particles can be advected for a month in seconds.. Maybe I am missing something?

Loading the global daily velocity fields substantially increased the computational costs of simulations. Hence, we load in the data between latitudes of 31°N and 31°S, to decrease the computational time of a simulation. After the particles travel for longer timescales, however, more particles pass these boundaries, giving the trajectories NaN values, in turn making the statistical analysis around the edges of the domain less robust as more trajectories become invalid. In the simulations, the particles were initially actually advected for 6 months, but for the reason above, we did not account for the data past the first month.

(c) line 150 :  Not clear how the correlation is calculated, please provide statistical details of the approach used. Moreover, why some metrics are correlated only with el Niño event and not la Niña?

Following also the comments from reviewer 2, we have decided to clarify the methods section as a whole. The metrics are now provided with a mathematical expression and graphic example, shown below in Fig. 1:

[Figure]

*Fig. 1 Conceptual visualisation of the metrics used in this study: displacement (top left), travel distance (top right), distance ratio (bottom left) and loopiness (bottom right)*

We added lines 154-158, to address how we calculate the correlations:

*"First, values from each metric (i.e., virtual particle's displacement, distance ratio, and loopiness) are grouped by the month in which the virtual particles were released. Second, a monthly average is calculated for each metric. Then, the correlation between the monthly average and the monthly Nino3.4 index is assessed using Pearson correlation statistics. Two-sided p-values are calculated under the assumption that the metrics and the Nino3.4 index are drawn from independent normal distributions."*

These p-values are used to indicate the areas where the calculated correlation values are not only high, but also statistically significant. As such, our metrics are correlated to the full Nino3.4 index, during both positive and negative ENSO events.

Following the feedback from reviewer 2, we revisited the method of the composite images (original manuscript Fig 2b, 2c), and replaced them with the same correlation analysis as the displacement and loopiness, shown below in Fig. 2a and 2b. The (a) correlation and (b) linear regression coefficient of the distance ratio of the depth-integrated virtual particles with the Niño3.4 index is shown below.

[Figure]

(a)

[Figure]

(b)

*Fig 2 **(a)** Correlation between virtual dFAD distance ratio after 1 month and the monthly Niño3.4 index. Positive correlation are shown in red, whereas blue colours show a negative correlation. The dotted black lines mark regions where correlations are significant (probability values larger than 95%). **(b)** Linear regression coefficient between virtual dFAD distance ratio after 1 month and the monthly Niño3.4 index. Units are in K$^{-1}$.*

**(d) line 156 : This mechanisms could be checked explicitly**

Following our response to point (3), we changed lines 156-158 (old version of the manuscript) to lines 172-176 (new version):

*"The pattern of alternating positive and negative correlations surrounding the NECC is likely due to the southward shift in the NECC during the 2009-2010 (Zhao et al., 2013) and 2015-2016 (Zhou et al., 2021) el Niño and the northward shift during la Niña. However, this latitudinal shift of the NECC does not occur during every ENSO event. Our analysis shows no significant correlations of distance ratio and loopiness in Fig. 3 and 4 in the NECC to explain these patterns with eddy-like behaviour."*

In the literature, we found that the latitudinal shift of the NECC occurs both during the 2009-2010 and 2015-2016 el Niño event, and we found that the MOi velocity data also captured this behaviour. We think stronger eddies near the NECC are less likely to be the cause, as this behaviour is not found in the distance ratio and loopiness analyses.

**(e) Line 170 : Also here, this hypothesis could be tested more explicitly**

Especially the strongly (anti-)correlated part in the western and central SEC is not a well-defined area in literature. We back our hypothesis that there is increased eddy-like behaviour in this region during el Niño compared to la Niña events, with the correlation analyses of the distance ratio and loopiness. Both metrics support the idea that eddy-like behaviour is enhanced in this region, though the underlying physical mechanisms are best suited for a separate study.

**Response to second reviewer**

*This paper assess the drift of fishing FADs in the tropical Pacific using a numerical model and a Lagrangian simulator for FAD particle trajectories. It is argued that the results of this study can help with sustainable management of FAD resources and FAD pollution. I found the paper difficult to understand because of the exceptionally poor quality of the presentation. I wrestled between rejection and major revision, eventually settling on rejection since I can't offer constructive comments on how to salvage such a poorly conceived and executed study. My comments follow.*

Below, we detail how we address the comments.

*1. The Mercator model product used for estimating ocean velocities is presented as a black box. There is no attempt to illustrate the ocean circulation in the model, including mean, seasonal cycle, and ENSO time scale variations. How does the model represent eddy variability, where is it highest and how does it change with ENSO cycle variations?*

*Also, there no attempt at model validation or even a summary of what others have to done to validate the model. There is no comparison of the surface flows with flows averaged over the upper 50 m, even though this is a major theme of the analysis. Instead, we are treated to a child-like drawing of the Pacific Ocean circulation in Figure A1.*

The reviewer is right that there was little validation of the hydrodynamic data that underpins the particle advection. We have now added a short summary of the validation of the MOI_GLO12_WEEKLY_run_for_DAILY_FORECAST product in the revised manuscript (lines 98-104):

"*This product has been validated globally in Lellouche et al (2013), where it is found that the hydrodynamic data agrees well with independent observations. A very recent publication (Fritz et al 2023) also confirms that the product captures the circulation in the Indonesian Throughflow and western Tropical Pacific well. However, it is known that the tropical SEC is slightly too strong in the MOi data when compared to in-situ 15m drogued drifters (Lellouche et al 2021), which may result in a slight overestimation of virtual dFAD displacement and loopiness for virtual particles released in the SEC. There is no dedicated validation of the full Tropical Pacific circulation, but our analysis shows that the Nino3.4 index in the hydrodynamic data agrees well with sea surface temperature observations (see below).*"

As this manuscript is a process study, we feel that this is sufficient validation of the hydrodynamic data.

*2. The definition of ENSO (Lines 141-144) is not conventional. Why do the authors use this instead or more conventional definitions? The higher threshold eliminates the 2006-07 El Nino which, though weak, is still considered an El Nino.*

Our initial line of thought was that isolating the strongest ENSO events would lead to features which are representative of the grown stages of an ENSO. We now agree with the reviewer that the threshold was poorly chosen and as such did the analysis again with a more conventional deviation threshold of at least 5 consecutive months, where the ONI index (3 month running mean of Niño3.4 temperature deviations) exceeds 0.5 °C. The 2006-2007 el Niño is still not a part in the current dataset, because it is not fully captured within the timeframe of the used dataset, but the 2018-2019 el Niño is.

Though this particular analysis is no longer used (explained in the response to point 3), this approach resulted in the figures below, Fig. 1 of our comments. These respectively show the updated version of Fig. 2b and 2c of the original manuscript, that showcase the average distance ratio over all el Niño and la Niña events.

[Figure]

a)

b)

*Fig. 1 Revised composite figures showing the average distance ratio of virtual dFADs after one month of travel during **(a)** all el Niño events and **(b)** all la Niña events. Values closer to 1 (in dark purple) indicate nearly straight trajectories, whereas values closer to 0 (in white) indicate stronger circulation. These figures are not used in the revised manuscript.*

3.  The average of two El Ninos, 2009-10 and 2015-16, as somehow representative of El Nino is not convincing or useful.  The two events are very different in spatial structure, amplitude, and temporal evolution.   The analysis is further flawed by not taking into account the time evolution of these events (lines 260-61), somehow assuming fixed circulation for only one point in time.

We partially disagree on the point that averaging the el Niño events is not useful, as we found that (with the new threshold values) the composite images of the 2009-2010, 2015-2016 and 2018-19 el Niño events are in general quite similar in their spatial structure of the calculated average distance ratio. These composite images are shown below in Fig. 2, but will not be a part of the revised manuscript. We agree that this method lacks the nuance of the temporal evolution of the individual events and as such chose to revisit this analysis altogether.

*2009-2010*

[Figure]

a)

*2015-2016*

[Figure]

b)

*2018-2019*

[Figure]

c)

*Fig. 2  Composite figures showing the average distance ratio of virtual dFADs after one month of travel during three separate el Niño events: (a) 2009-2010; (b) 2015-2016; (c) 2018-2019.*

*Values closer to 1 (in dark purple) indicate nearly straight trajectories, whereas values closer to 0 (in white) indicate stronger circulation. These figures are not used in the revised manuscript.*

To address the issues mentioned above, the same correlation analysis for this metric as was done for the other metrics is performed, which we agree gives a better representation of the ENSO variability. The (a) correlation and (b) linear regression coefficient of the distance ratio of the depth-integrated virtual particles with the Niño3.4 index is shown below, in Fig. 3. These two figures will be included in the new version and will replace Fig. 2b and 2c of the original manuscript.

[Figure]

(a)

(b)

*Fig 3. **(a)** Correlation between virtual dFAD distance ratio after 1 month and the monthly Niño3.4 index. Positive correlations are shown in red, whereas blue colours show a negative correlation. The dotted black lines mark regions where correlations are significant (probability values larger than 95%). **(b)** Linear regression coefficient between virtual dFAD distance ratio after 1 month and the monthly Niño3.4 index. Units are in $K^{-1}$.*

4. The written descriptions of displacement distance, travel distance, distance ratio, and "loopiness" are insufficient. These concepts should be illustrated with concrete graphic examples to make them understandable.

This is a good suggestion by the reviewer. We now include a graphic to illustrate the metrics better, shown in Fig. 4, as well as corresponding mathematical expressions in the textual description.

[Figure]

*Fig. 4  Conceptual visualisation of the metrics used in this study: displacement (top left), travel distance (top right), distance ratio (bottom left) and loopiness (bottom right)*

(a) Lines 43-45.  The description of the currents is incorrect.  The flows are not in the directions indicated but in the opposite directions.

We have corrected this.

(b) Lines 76-77.  35 to 65,000 or 35,000 to 65,000?

We have corrected this to 35,000.

(c) Lines 95-96.  Why did you use this period of time?

This was the time window for which MOi velocity-field data were available when we were running the particle simulations for this research.

**List of relevant changes**

Minor changes for legibility and conciseness purposes (e.g., 'These' instead of 'The', addition of 'virtual dFADs', removal of redundant subordinate clauses) will not be listed separately. These are marked in the file with the tracked changes. Line numbers are given according to the new manuscript version – these do not match the line numbers in the track changes document.

*Abstract*
- Minor changes for legibility and conciseness.

*Introduction*
- Line 74: following comment (b) by reviewer #2, we changed 35 into 35,000.
- Line 74-81: this paragraph has been restructured and expanded upon for clarity. To justify our use of Lagrangian simulations, we cite additional references to previous research that uses Lagrangian simulations to study the drift of virtual particles, such as Amemou et al. (2020) and Curnick et al. (2021).

*Methods*
- Line 98-104: following comment 1 from reviewer #2, we added a paragraph on the validation of the MOI_GLO12_WEEKLY_run_for_DAILY_FORECAST product in existing research.
- Line 117-120: following comment 2 from reviewer #1, we substantiated our method, in which we do not account for mass and small scale hydrodynamic properties of (virtual) dFADs.
- Line 129: renamed the subsection 'Variables' to 'Metrics'.
- Following comment 4 by reviewer #2, the following changes were made:
  - line 135-136 added a mathematical expression for travel distance;
  - line 139-140 added a mathematical expression for distance ratio;
  - line 147-148 added a mathematical expression for loopiness
  - figure 1 is added as a new figure to visualize the metrics.
- Line 153-160: following comment (c) from reviewer #1 and comment 4 from reviewer #2, we overhauled our description of the correlation analysis and changed part of the method altogether.

*Results*
- Line 172-176: following comment (d) from reviewer #2, we link the pattern of alternating positive and negative correlations in the NECC of the distance ratio to the latitudinal shift of the NECC associated with ENSO, as found in literature.
- Line 180-189: following comment 3 from reviewer #2, the composite images have been replaced with a correlation analysis (described in more detail above). Using this improved method, we now analyse different figures in this section.
- Figure 3 (previously Figure 2) now shows the correlation analysis (correlation & regression coefficient) of the distance ratio instead of the composite images, including a new caption.

- Line 206-209: following comment 3 from reviewer #1, we added sources to connect the loopiness patterns observed in the northern EPO to ENSO variability of tropical instability waves and -vortices.
- Figure 5c and 5d now show the new correlation analysis of the distance ratio.
- Line 218-220: since figures 5c and 5d have changed, we describe our improved results.
- Line 224: we added a more clear conclusion to this paragraph, stating that we observe stronger variability of dFAD drift with ENSO in the case of surface-only flow.

*Discussion and outlook*
- Line 260: removed the sentence about the shift of FAD density in the southern WCPO during el Niño, due to lack of verification.
- Line 266-273: following comment 3 from reviewer #1, we expanded the discussion on the connection between loopiness/ distance ratio and physical mechanisms found in literature.
- Line 277: following comment 2 from reviewer #2, **three** el Niño events occur during our simulation instead of two, due to a more commonly used definition of an ENSO event.
- Line 298 (code availability): changed to new doi of the model.